# Possibility of Exciton Bose–Einstein Condensation in CdSe Nanoplatelets

**DOI:** 10.3390/nano13192734

**Published:** 2023-10-09

**Authors:** Davit A. Baghdasaryan, Volodya A. Harutyunyan, Eduard M. Kazaryan, Hayk A. Sarkisyan, Lyudvig S. Petrosyan, Tigran V. Shahbazyan

**Affiliations:** 1Institute of Engineering and Physics, Russian-Armenian University, H. Emin 123, Yerevan 0051, Armenia; davit.baghdasaryan@rau.am (D.A.B.); volodya.harutyunyan@rau.am (V.A.H.); eduard.ghazaryan@rau.am (E.M.K.); 2Institute of Electronics and Telecommunications, Peter the Great Saint-Petersburg Polytechnical University, 195251 Saint-Petersburg, Russia; 3Department of Physics, Jackson State University, Jackson, MS 39217, USA; ludvig1977@gmail.com (L.S.P.); shahbazyan@jsums.edu (T.V.S.)

**Keywords:** nanoplatelets, size quantization, exciton states, Bose–Einstein condensation

## Abstract

The quasi-two-dimensional exciton subsystem in CdSe nanoplatelets is considered. It is theoretically shown that Bose–Einstein condensation (BEC) of excitons is possible at a nonzero temperature in the approximation of an ideal Bose gas and in the presence of an “energy gap” between the ground and the first excited states of the two-dimensional exciton center of inertia of the translational motion. The condensation temperature ($T_{c}$) increases with the width of the “gap” between the ground and the first excited levels of size quantization. It is shown that when the screening effect of free electrons and holes on bound excitons is considered, the BEC temperature of the exciton subsystem increases as compared to the case where this effect is absent. The energy spectrum of the exciton condensate in a CdSe nanoplate is calculated within the framework of the weakly nonideal Bose gas approximation, considering the specifics of two-dimensional Born scattering.

## 1. Introduction

Various ultra-thin atomically flat quasi-two-dimensional semiconductor nanostructures with planar geometry known as nanoplatelets (NPLs) have been at the center of intense research in the last decade (see, e.g., Refs. [1,2,3,4,5,6,7,8,9,10,11,12] and references therein). Such great interest in these structures is because, on the one hand, semiconductor NPLs have the properties of semiconductor nanocrystals of the previous generation that are already widely used in electronics and optoelectronics and a wide range of other applications: quantum dots (QDs) [13,14,15] and nanorods (NRs) [16]. While, on the other hand, they exhibit several completely new unique properties that are absent not only in a bulk sample but also in the listed nanostructures of the previous generation. In particular, the giant oscillator strength, narrow emission, large absorption cross-section, sizeable optical gain, and a much higher binding energy of a two-dimensional exciton as was illustrated compared to a case involving quantum film that had been observed in NPLs [1,4,5,17,18,19,20]. Therefore, semiconductor NPLs are very promising materials that can be used to create several optoelectronic devices: light emitter devices of various ranges, light generation and amplification devices [12,16], lasers [16,17,18], solar energy harvesting applications [19,20], photodetectors [21,22,23,24], photosensors [22,23,24], etc. Note that the exceptional fluorescent properties of NPLs make them promising nanomaterials for use in biological and medical applications [9,25,26]. NPLs based on **II**–**VI** compounds (CdS, CdSe, CdTe [1,2,3,4,5,6,7,8,9,10,11,12,16,17,18,19,20,21,22,23,24,25,26,27,28], HgS, HgSe, HgTe [29,30,31]) and, in part, on **IV**–**VI** compounds (PbS, PbSe, PbTe [32,33]) are actively investigated. In what follows, our attention will be focused on NPLs of CdSe.

The intensive investigation of nanoplatelets (NPLs) is primarily motivated by the distinct physical properties arising from the unique characteristics of single-particle states and the statistical ensemble of charge carriers within the nanocrystal. Furthermore, at high excitation levels, the excitonic states and collective properties of excitons, as well as those of exciton complexes such as trions and biexcitons, also contribute to the observed phenomena [34,35,36]. The size, material composition, structural attributes of the NPLs, and the surrounding environment all play crucial roles in determining the formation and physical characteristics of these states.

As previously highlighted, one of the most remarkable manifestations of the distinctive physical characteristics exhibited by quasi-two-dimensional NPLs, exemplified by colloidal CdSe NPLs, resides in the significantly elevated binding energy of quasi-two-dimensional excitons when juxtaposed with not only their bulk counterparts (approximately ~15 meV) but also in comparison to “conventional” quantum wells (approximately ~60 meV and a range of 190–200 meV and beyond) [37,38]. The substantial binding energy, consequently endowing CdSe NPL excitons with relatively robust stability, not only affords the prospect of discerning exceptionally narrow absorption and emission spectral peaks and exceptionally brief radiative fluorescence lifetimes (less than 1 nanosecond) [4,39] but also engenders the capacity to explore the collective characteristics of excitons within these NPL structures at elevated excitation levels [40,41]. Wannier–Mott excitons within semiconductors, including CdSe NPLs, are composite entities, designated as bosons, which theoretically possess the potential for Bose–Einstein condensation (BEC). Investigations into the feasibility of BEC occurrence among semiconductor excitons commenced in the 1960s and have been actively pursued (as evidenced by comprehensive reviews [42,43] and their accompanying references). It is noteworthy, however, that amidst the wealth of persuasive theoretical inquiries, empirical experiments have thus far refrained from providing definitive confirmation of exciton BEC. Concurrently, we take cognizance of recent publications (such as Ref. [44]), which have once again reported the experimental observation of exciton BEC in bulk semiconductors. The trend towards the miniaturization of semiconductor-based devices and the consequent proliferation of low-dimensional semiconductor technologies naturally arouses curiosity regarding the collective attributes, with a particular emphasis on the phenomenon of exciton BEC, within low-dimensional semiconductor structures, including quasi-two-dimensional structures such as NPLs.

It is well known that BEC is not possible in an ideal two-dimensional system of bosons [42,43,45]. However, due to the presence of various “internal” physical factors (defects, disorder, localizing wells, etc.) and external modulating influences (external fields, size quantization, traps), almost always in quasi-two-dimensional semiconductors there are real deviations of the sample properties from ideal two-dimensionality. So, under certain physical conditions, BEC of excitons can also become possible in quasi-two-dimensional semiconductors (see, for example, Refs. [43,46,47,48,49,50,51,52,53,54,55,56,57]). In particular, the possibility of BEC of excitons in CdS and CdSe quantum semiconductor films with the wurtzite structure was theoretically considered in [46]. The specific feature that makes it possible to observe the BEC of excitons in these quasi-two-dimensional structures is the deviation of the dispersion law of the translational motion of excitons in the film plane from strictly quadratic. The possibility of exciton BEC in a semiconductor quantized disk [47] is due to the discreteness of the energy spectrum of the translational motion of excitons in the disk. In Refs. [48,50,55], the role of potential traps in a sample for the onset of exciton BEC in quasi-two-dimensional semiconductors was demonstrated, while, in [51], the possibility of exciton BEC in an ideal two-dimensional system in the presence of a strong transverse magnetic field was considered. The possibility of BEC of excitons in a quasi-two-dimensional structure in the presence of a disorder in the energy spectrum of excitons was studied in [43,54]. In [56], the BEC of interlayer two-dimensional excitons in atomically flat two-layer semiconductors was considered, and, in [57], it was shown that in a quasi-two-dimensional sample, for the onset of BEC of excitons, along with their high binding energy and long lifetime compared to thermalization time, the symmetry of the zones of the structure under consideration is also significant. Accounting for the latter leads, in fact, to an increase in the calculated BEC temperature.

The significant advantage of NPLs is the perfection of geometric shapes and geometric shape variations compared to core-shell QDs. In addition to what has been said, it should be noted that, in NPLs, dielectric effects are vividly manifested due to the presence of a dielectric environment; as a result, the NPL binding energies reach large values compared to spherical core-shell quantum dots. Also, the exciton subsystem becomes more stable.

This study delves into the theoretical exploration of BEC of two-dimensional excitons within the framework of an ideal Bose gas model, encompassing the interplay of screening effects between excitons and unbound charge carriers within a CdSe NPL. The central focus of our investigation entails the computation of the energy spectrum considering the weak nonideality of the exciton subsystem. It should be mentioned that, in the present paper, we discuss the equilibrium excitonic system and do not consider some of the non-equilibrium optical effects. Particularly we do not discuss the effect of fluorescent emission in CdSe under the influence of external radiation.

The structure of the article is as follows: In Section 1, within the framework of the ideal Bose gas model, the possibility of BEC of two-dimensional excitons in a CdSe NPL is considered. In Section 2, the theoretical approach for Bose–Einstein condensation of excitons in the frameworks of ideal (Section 2.1 and Section 2.2) and weakly nonideal Bose gas (Section 2.3) is presented. In Section 3, the results obtained in the work are discussed and, in Section 4, the conclusion related to the work is presented. In Appendix A, the diagonalization of Hamiltonian is presented.

## 2. Theory

### 2.1. Exciton Subsystem in CdSe NPL in the Ideal Bose Gas Approximation

The Bohr radius of an exciton in bulk CdSe crystal ($a_{ex}^{3D}$) is of the order of 5–6 nm [38]. When implementing the weak quantization regime in the CdSe NPL ($L_{x}\times L_{y}$) plane:(1)$$L_{x},L_{y}\gg a_{ex}^{3D}$$
where $L_{x},L_{y}$ reaches tens and even several hundreds of nanometers [38]; in the NPL plane, an electron, and a hole are bound into a two-dimensional Wannier–Mott exciton. The energy of the Wannier–Mott exciton, as a composite particle, in CdSe NPL in the weak quantization mode is written in the following form:(2)$$E_{tot}^{ex}=E_{ex,∥}^{conf}+E_{bind}^{2D}+E_{e,⊥}^{conf}+E_{h,⊥}^{conf}\equiv E_{n_{x},n_{y}}^{conf}+E_{b,n}^{2D}+E_{n_{e}}^{conf}+E_{n_{h}}^{conf}$$

Here, $E_{ex,∥}^{conf}\equiv E_{n_{x},n_{y}}^{conf},E_{bind}^{2D}\equiv E_{b,n}^{2D},E_{e,⊥,}^{conf}\equiv E_{n_{e}}^{conf},E_{h,⊥}^{conf}\equiv E_{n_{h}}^{conf}$ are the energy of the quantized motion of the center of mass of an exciton in the CdSe NPL plane, the binding energy of a two-dimensional exciton, and the quantization energies of an electron and a hole along the z-axis, respectively. To calculate specifically each of these energies, we will use the values of physical quantities and the corresponding approximations presented in Ref. [38], where the theoretical calculations performed are in sufficient agreement with the experimental results.

In CdSe NPLs, the exciton thermalization (cooling) time ($\tau_{T}\sim 10fs-1ps$) is much shorter than their recombination time ($\tau_{rec}\sim 100ps-10ns$) [6] (see Figure 1d in [6]). This circumstance makes it possible to consider the statistical features of the exciton subsystem at high levels of nanocrystal excitation; in particular, the possibility of exciton BEC in CdSe NPL.

Let us first consider the simplest case, where it is assumed that all electrons and holes that appear as a result of the implementation of transitions at high excitation levels are bound in pairs into two-dimensional excitons and the interaction between excitons can be neglected. It is also assumed that there is no effect of any external fields on excitons.

It was shown in [47,58,59] that BEC can be realized in an ideal Bose gas with a completely discrete energy spectrum of particles at a nonzero temperature. In the case under consideration, for the total number of particles (N) of the equilibrium exciton subsystem, we have:(3)$$N=g_{s}{\sum_{\left\{k\right\}}\left[\exp\left(\frac{E_{tot,k}^{ex}-\mu }{k_{B}T}\right)-1\right]}^{-1}$$

Here, $g_{s}$—spin degeneracy factor of the two-dimensional exciton, $\left\{k\right\}\;-$ a set of quantum numbers that determine the total energy of an exciton (2) in a given state, $k_{B}-$ Boltzmann’s constant, T—Temperature, μ—chemical potential of the equilibrium exciton subsystem. It is clear from expression (3) that the following conditions must necessarily be satisfied for the chemical potential:(4)$$\mu <E_{tot}^{ex};\;\;\mu_{\max}={\left(E_{tot}^{ex}\right)}_{\min}={\left(E_{ex,∥}^{conf}\right)}_{\min}+{\left(E_{bind}^{2D}\right)}_{\min}+{\left(E_{e,⊥}^{conf}\right)}_{\min}+{\left(E_{h,⊥}^{conf}\right)}_{\min}$$

Let us single out, from expression (3), the total number of particles N and those terms that make up the number of particles $N^{\prime }$ at excited energy levels $E_{tot,k}^{ex}>{\left(E_{tot,k}^{ex}\right)}_{\min}$:(5)$$N^{\prime }(T,\mu )=g_{s}{\sum_{\left\{k\right\}>{\left\{k\right\}}_{\min}}\left[\exp\left(\frac{E_{tot,k}^{ex}-\mu }{k_{B}T}\right)-1\right]}^{-1}$$

Considering conditions (4), it is easy to conclude that (a) series (5) converges at any final temperature and (b) at any considered temperature T. At the same time, the expression below:(6)$$N^{*}\left(T,\mu_{\max}\right)=g_{s}{\sum_{\left\{k\right\}>{\left\{k\right\}}_{\min}}\left[\exp\left(\frac{E_{tot,k}^{ex}-\mu_{\max}}{k_{B}T}\right)-1\right]}^{-1}$$
will be the upper limit for series (5):(7)$$N^{\prime }\left(T,\mu \right)\leq N^{*}(T,\mu_{\max})$$

Therefore, in the system under consideration, the number of particles at excited levels at any temperature is finite and is bounded from above, and this upper limit decreases with decreasing temperature, turning to zero in the limit of zero temperature. Moreover, the condition $\;\mu_{\max}={\left(E_{tot}^{ex}\right)}_{\min}$ is satisfied already at some non-zero temperature $T=T_{c}\neq 0$, which will be the BEC temperature in the system under consideration. We calculate the temperature $T_{c}$ using the condition that the total number of particles at $T=T_{c}$ and $\;\mu_{\max}={\left(E_{tot}^{ex}\right)}_{\min}$ is equal to the number of particles at excited levels at the same values of temperature and chemical potential:(8)$$N=N^{*}\left(T_{c},\mu_{\max}\right)=g_{s}{\sum_{\left\{k\right\}>{\left\{k\right\}}_{\min}}\left[\exp\left(\frac{E_{tot,k}^{ex}-\mu_{\max}}{k_{B}T_{c}}\right)-1\right]}^{-1}.$$

In the case where condition (1) is satisfied, the energy of the quantized motion of the center of mass of the exciton $E_{ex,∥}^{conf}$ is much less than the energies $E_{bind}^{2D},E_{e,⊥}^{conf},E_{h,⊥}^{conf}$. Based on this, when summing over energy states $E_{ex,∥}^{conf}=E_{n_{x},n_{y},}^{conf}$ we can restrict ourselves to considering only the first two levels of size quantization without distorting the physical essence of the problem. The remaining excited states of the two-dimensional exciton center of mass motion in the XY plane can be calculated in the semiclassical approximation. Considering the above, from the normalization condition (3), in order to determine the dependence of the chemical potential of the system μ on the two-dimensional exciton density $n^{2D}=N/S,\left(S=L_{x}\times L_{y}\right)$, we arrive at the following relationship:(9)$$\begin{array}{l} n^{2D}=\sum_{n_{e},n_{h}}\sum_{n}\frac{1}{S}{\left[\exp\left(\frac{E_{n_{e},n_{h}}^{conf}+{\left(E_{ex,∥}^{conf}\right)}_{\min}+E_{b,n}^{2D}-\mu }{k_{B}T}\right)-1\right]}^{-1}-\\ -g_{s}\frac{Mk_{B}T}{2\pi \hbar^{2}}\sum_{n_{e},n_{h}}\sum_{n}\ln\left|1-\exp\left(\frac{\mu -E_{n_{e},n_{h}}^{conf}-{\left(E_{ex,∥}^{conf}\right)}_{2}-E_{b,n}^{2D}}{k_{B}T}\right)\right| \end{array}$$

Here, $E_{n_{e},n_{h}}^{conf}=E_{n_{e}}^{conf}+E_{n_{h}}^{conf}$ is the total energy of size quantization of an electron and a hole along the z-axis. For simplicity in calculations, we restrict ourselves to the case $L_{x}=L_{y}=L$. Then,
(10)$${\left(E_{ex,∥}^{conf}\right)}_{\min}\equiv {\left(E_{ex,∥}^{conf}\right)}_{1}=E_{n_{x}=1,n_{y}=1}^{conf}=2\frac{\pi^{2}\hbar^{2}}{2ML^{2}};\;{\left(E_{ex,∥}^{conf}\right)}_{2}=2\frac{\pi^{2}\hbar^{2}\cdot 5}{2ML^{2}};\;M=m_{∥}^{e}+m_{∥}^{h}$$

The summation in expression (9) is carried out over the quantum numbers of all energy states, starting from the ground state.

Figure 1 shows the dependence of the chemical potential μ of an equilibrium system of excitons in CdSe NPL on the two-dimensional density of particles $n^{2D}$ at various values of the system temperature T for system dimensions in the plane $L=L_{x}=L_{y}=25\;\mathrm{nm}$ (a), $L=L_{x}=L_{y}=100\;\mathrm{nm}$ (b).

Figure 1a,b clearly show that in the considered exciton subsystem, depending on the density of particles, the chemical potential actually reaches its maximum value, albeit at different, but necessarily nonzero, temperatures.

The corresponding values of the physical quantities characterizing the structure under consideration, for which the calculations of the graphs in Figure 1a,b, according to expressions (3) and (9), are given in Table 1.

With the given parameters, for the largest value of the chemical potential from expression (4), we have at $L_{x}=L_{y}=100\;\mathrm{nm}$ $\mu_{\max}\approx 0.408\;\mathrm{eV}$ and at $L_{x}=L_{y}=25\;\mathrm{nm}$ $\mu_{\max}\approx 0.413\;\mathrm{eV}$. Also, with an increase in the energy of the fundamental level of size quantization of the translational motion of the center of mass of a two-dimensional exciton, the limiting value of the chemical potential of the exciton subsystem increases.

The general expression (8) for determining the condensation temperature $T_{c}$ in the case under consideration is reduced to the following transcendental relation:(11)$$\begin{array}{l} n^{2D}=\frac{1}{S}\sum_{n_{e},n_{h}}\sum_{n}{\left[\exp\left(\frac{E_{n_{e},n_{h}}^{conf}+{\left(E_{ex,∥}^{conf}\right)}_{\min}+E_{b,n}^{2D}-\mu_{\max}}{k_{B}T_{c}}\right)-1\right]}^{-1}-\\ -g_{s}\frac{Mk_{B}T_{c}}{2\pi \hbar^{2}}\sum_{n_{e},n_{h}}\sum_{n}\ln\left|1-\exp\left(\frac{\mu_{\max}-E_{n_{e},n_{h}}^{conf}-{\left(E_{ex,∥}^{conf}\right)}_{2}-E_{b,n}^{2D}}{k_{B}T_{c}}\right)\right| \end{array}$$

In the first term, the summation over the energy states is carried out over all excited states, and, in the second term, over all states, starting from the states with the minimum energy.

Figure 2 shows the dependence of the condensation temperature $T_{c}$ on the value of the two-dimensional particle density $n^{2D}$.

As can be seen, at the same particle density, with a decrease in the longitudinal dimensions of the NPL, the condensation temperature rises in all the cases.

### 2.2. Degenerate Exciton Subsystem in CdSe NPL in the Presence of Screening

In real samples, as is known, certain deviations from ideality in the exciton subsystem are present, and, in some cases, it becomes necessary to consider the nonideality factor. A similar situation arises, for example, in the case of a stable exciton subsystem created in CdSe NPL at high levels of photoexcitation. In this case, along with the exciton subsystem, the sample also contains a subsystem of unbound electrons and holes, as a result of which free charges screening effects on the properties of bound two-dimensional excitons begin to manifest themselves significantly [40,41]. However, we note from the very beginning that, on the one hand, this screening is so significant that it affects the binding energy of the two-dimensional exciton, but, at the same time, it is not so strong as to break up the bound state of the exciton and transform it into an electron-hole continuum [41].

According to the conclusions of the authors of [40,41], in the experiments performed in CdSe NPL, a quantum-degenerate exciton gas is realized, which opens the prospect of observing superfluidity in these nanostructures. Naturally, one of the main steps in revealing the possibility of superfluidity manifestation is the elucidation of the possibility of Bose condensation of excitons under these physical conditions.

For a quantitative description of the exciton subsystem, in this case, the energies $E_{ex,∥}^{conf}\equiv E_{n_{x},n_{y}}^{conf},E_{e,⊥,}^{conf}\equiv E_{n_{e}}^{conf},E_{h,⊥}^{conf}\equiv E_{n_{h}}^{conf}$ remain the same both in meaning and values as in Section 2. Due to the screening of free charges in the NPL, the binding energy of the two-dimensional exciton $E_{bind}^{2D}\equiv E_{b,n}^{2D}$ changes. It was shown in [41] that the binding energy of a two-dimensional exciton in the presence of screening of free charges is determined by the density of the charges $n_{q}$ or, ultimately, by the photopump density $n_{\gamma }$. Accordingly, in this section, we will denote the binding energy of a two-dimensional exciton by $E_{b,n}^{2D}\left(n_{q}\right)$. According to the results in [41], the binding energy of the ground state of a two-dimensional exciton in the presence of screening varies from ${\left[E_{b,n}^{2D}\left(n_{q}\right)\right]}_{\min}=\left[E_{b,1}^{2D}\left(n_{q}=0\right)\right]=-193\;\mathrm{meV}$, which corresponds to the absence of screening, to the value ${\left[E_{b,n}^{2D}\left(n_{q}\right)\right]}_{sut}=\left[E_{b,n=1}^{2D}\left(n_{q}\to n_{q}^{\infty }\right)\right]=-45\;\mathrm{meV}$, which corresponds to the maximum possible screening action, after which the screening effect of free charges on excitons saturates. However, excitons remain stable, meaning that, in this case, it is expedient to consider the properties of the exciton subsystem under degeneracy conditions in the limiting cases $n_{q}\to 0$ and $n_{q}\to n_{q}^{\infty }$, when the exciton binding energy is practically constant or when it is not subjected to the screened action of free charges and practically does not depend on the temperature of the system [41].

Let us now turn to the consideration of the properties of the degenerate exciton subsystem in the limiting cases of absence $n_{q}\to 0$ and saturation $n_{q}\to n_{q}^{\infty }$ of the screened effect of free charges on the exciton binding energy in CdSe NPL.

By analogy with Equation (11), to determine the critical temperature in this case, we have the following condition:(12)$$\begin{array}{l} n_{ex}^{2D}=\frac{1}{S}\sum_{n_{e},n_{h}}{\left[\exp\left(\frac{E_{n_{e},n_{h}}^{conf}+{\left(E_{ex,∥}^{conf}\right)}_{\min}+E_{b,2}^{2D}-\mu_{\max}}{k_{B}T_{c}}\right)-1\right]}^{-1}-\\ -g_{s}\frac{Mk_{B}T_{c}}{2\pi \hbar^{2}}\sum_{n_{e},n_{h}}\sum_{n=1,2}\ln\left|1-\exp\left(\frac{\mu_{\max}-E_{n_{e},n_{h}}^{conf}-{\left(E_{ex,∥}^{conf}\right)}_{2}-E_{b,n}^{2D}}{k_{B}T_{c}}\right)\right| \end{array}$$

In this expression, when summing over the “internal” exciton energy levels, we restrict ourselves to only the first two levels:

In the case $n_{q}\to 0$, there is no screening.

According to [41], for the energy levels of a two-dimensional exciton we have:

${\left[E_{b,n}^{2D}\left(n_{q}\right)\right]}_{\min}=\left[E_{b,1}^{2D}\left(n_{q}=0\right)\right]=-$193 meV, $\left[E_{b,2}^{2D}\left(n_{q}=0\right)\right]=-40$ meV.

In this case, the behavior of the condensation temperature of an ideal gas coincides with the behavior shown in Figure 2.

In the case $n_{q}\to n_{q}^{\infty }$, the screening reaches saturation.

Under these conditions, for the energy levels of a two-dimensional exciton, we have [41]:${\left[E_{b,n}^{2D}\left(n_{q}\right)\right]}_{sut}=\left[E_{b,1}^{2D}\left(n_{q}\to n_{q}^{\infty }\right)\right]=-45$ meV, $\left[E_{b,2}^{2D}\left(n_{q}\to n_{q}^{\infty }\right)\right]=-12$ meV.

Figure 3 shows the curves of the condensation temperature on the density of the particles in conditions where the screening reaches saturation.

### 2.3. The Degenerate Exciton Subsystem in CdSe NPL in the Approximation of a Weakly Nonideal Bose Gas

Following the technique developed for the three-dimensional case [45], we also consider the degenerate quasi-two-dimensional exciton subsystem in CdSe NPL in the approximation of a weakly nonideal Bose gas (ignoring spin). The basis for the application of this method, as is known, is the Born approximation [45]. For two-dimensional exciton-exciton scattering, the Born approximation is applicable if the repulsion potential between particles satisfies the following condition [60,61]:(13)$$\left|U\left(\vec{\rho }\right)\right|\ll \frac{\hbar^{2}\ln^{-1}\left(2e^{-C}/K\rho_{0}\right)}{2M\rho_{0}^{2}};\;K=\sqrt{2ME/\hbar^{2}}$$

Here, $\rho_{0}$ is the region of action radius of effective filed $U\left(\vec{\rho }\right)$; E the energy of the scattered particle. At the same time, the condition of weak nonideality can be formulated as the condition of smallness of the range radius $\rho_{0}$ of the repulsive potential $U\left(\vec{\rho }\right)$ in comparison with the average distance between particles $l_{0}:$(14)$$\rho_{0}\ll l_{0}\sim {\left(N/S\right)}^{1/2}$$

In the language of momenta, this condition is equivalent to the proposition that the momenta of scattering particles are small [45]:(15)$$p\rho_{0}/\hbar \ll 1$$

Physically, this means that the de Broglie wavelength of the scattered particles in this case is much larger than the size of the region of action of the field $U\left(\vec{\rho }\right)$.

In the case of an ideal gas, at a temperature below the condensation temperature $\left(T<T_{c}\right)$, all particles of the gas will be in the condensate: $N_{0}=N,\;N^{\prime }=0$. If there is a weak repulsion between the particles, scattering acts will take place and, as a result, some particles may end up in an upper condensate energy state also at $T<T_{c}:$(16)$$N^{\prime }\neq 0,N^{\prime }\ll N_{0}∼N$$

In the representation of the second quantization, the Hamiltonian of such states, minus the size quantization energy of particles and in considering only the pair interaction between particles, the general case is represented in the following form:(17)$$H=\sum_{\vec{p}}\left[\frac{p^{2}}{2M}\right]a_{\vec{p}}^{+}a_{\vec{p}}+\frac{1}{2}\sum_{{\vec{p}}_{1},{\vec{p}}_{2}}\sum_{{\vec{p^{\prime }}}_{1},{\vec{p^{\prime }}}_{2}}\langle {\vec{p^{\prime }}}_{1},{\vec{p^{\prime }}}_{2}|U\left(\vec{\rho }\right)|{\vec{p}}_{1},{\vec{p}}_{2}\rangle a_{{\vec{p^{\prime }}}_{1}}^{+}a_{{\vec{p^{\prime }}}_{2}}^{+}a_{{\vec{p}}_{1}}a_{{\vec{p}}_{2}}$$

Here, $a_{\vec{p}}^{+},a_{\vec{p}}$—operators of creation and annihilation of particles (bosons), $\vec{p},\vec{p^{\prime }}$—the momenta of the particle before and after scattering, and the index 2 is the band index describing the size quantization of particle motion in the CdSe NPL plane when the particle is in the over condensate state as a result of scattering. Considering the condition of small momenta, the general matrix element in (17) can be replaced by its value at zero momentum [45]:(18)$$\langle {\vec{p^{\prime }}}_{1},{\vec{p^{\prime }}}_{2}|U\left(\vec{\rho }\right)|{\vec{p}}_{1},{\vec{p}}_{2}\rangle \to \langle {0^{\prime }}_{1},{0^{\prime }}_{2}|U\left(\vec{\rho }\right)|0_{1},0_{2}\rangle \equiv \frac{U_{0}}{S}$$

In using the calculations given in [45] regarding the order of expressions containing products of quantities $a_{p}^{+},a_{p}$ and considering the law of conservation of momentum and approximation (18) instead of (17) for the Hamiltonian of the system, we obtain:(19)$$H=\frac{N^{2}}{2S}U_{0}+\sum_{\vec{p}}\left[\frac{{\vec{p}}^{2}}{2M}\right]a_{\vec{p}}^{+}a_{\vec{p}}+\frac{N}{2S}U_{0}\sum_{\vec{p}\neq 0}\left(a_{\vec{p}}a_{-\vec{p}}+a_{\vec{p}}^{+}a_{-\vec{p}}^{+}+2a_{\vec{p}}^{+}a_{\vec{p}}\right)$$

The matrix element of exciton–exciton scattering included in (19) must be expressed in terms of the parameters describing the scattering process itself. For two-dimensional scattering, the scattering amplitude in the Born approximation is given by the following expression [61]:(20)$$f=-\frac{M}{\hbar^{2}\sqrt{2\pi K}}\int U\left(\vec{\rho }\right)\exp\left(-i\vec{q}\cdot \vec{\rho }\right)d\vec{\rho },$$
where $\vec{q}$ is the transmitted impulse. Considering the small momenta of the particles involved in the scattering event, we can set $e^{-i\vec{q}\cdot \vec{\rho }}\approx 1$, after which we will have the following for the scattering amplitude:(21)$$f\approx -\frac{M}{\hbar^{2}\sqrt{2\pi K}}\int U\left(\vec{\rho }\right)\;d\vec{\rho }=-\frac{MU_{0}}{\hbar^{2}\sqrt{2\pi K}}.$$

As follows from (21), in the considered case, the scattering amplitude in the Born approximation is not a constant value, but it is inversely proportional to $\sqrt{K}$:(22)$$f\left(E\right)\sim E^{-1/4}.$$

However, in the case under consideration, the following dimensionless quantity will be constant:(23)$$\alpha =-\sqrt{K}f\left(K\right)$$

In the first and second Born approximations, the matrix element $U_{0}$ will be expressed in terms of the constant α as follows:

First Born approximation:(24)$$U_{0}=\frac{\hbar^{2}\sqrt{2\pi }}{M}\alpha .$$

Second Born approximation:(25)$$U_{0}=\frac{\hbar^{2}\sqrt{2\pi }}{M}\alpha \left[1-\frac{\hbar^{2}\sqrt{2\pi }}{MS}\alpha \sum_{{{\vec{p}}^{\prime }}_{1}}\frac{2M}{{\vec{p}}_{1}^{2}+{\vec{p}}_{2}^{2}-{{\vec{p}}^{\prime }}_{1}^{2}-{{\vec{p}}^{\prime }}_{2}^{2}}\right].$$

Considering that the collision takes place between the particles of the condensate, i.e., in this case:(26)$${\vec{p}}_{1}={\vec{p}}_{2}=0,\;\;{{\vec{p}}^{\prime }}_{1}=-{{\vec{p}}^{\prime }}_{2}\equiv \vec{p}$$

From having (23)–(25) for the Hamiltonian (19), we get:(27)$$\begin{array}{l} H=\sum_{\vec{p}}\left[\frac{{\vec{p}}^{2}}{2M}\right]a_{\vec{p}}^{+}a_{\vec{p}}+\frac{\hbar^{2}\sqrt{2\pi }}{2M}\alpha \frac{N^{2}}{S}\left(1+\frac{\hbar^{2}\sqrt{2\pi }}{S}\alpha \sum_{\vec{p}\neq 0}\frac{1}{{\vec{p}}^{2}}\right)+\\ \frac{\hbar^{2}\sqrt{2\pi }}{2M}\alpha \frac{N}{S}\sum_{\vec{p}\neq 0}\left(a_{\vec{p}}a_{-\vec{p}}+a_{\vec{p}}^{+}a_{-\vec{p}}^{+}+2a_{\vec{p}}^{+}a_{\vec{p}}\right) \end{array}$$

After diagonalizing the Hamiltonian (27), we arrive at the following result (see Appendix A):(28)$$\begin{array}{l} H=E_{0}+\sum_{\vec{p}\neq 0}\varepsilon \left(p\right)b_{\vec{p}}^{+}b_{\vec{p}};\;\;E_{0}=\frac{1}{2}\sum_{\vec{p}\neq 0}\left\{\varepsilon \left(p\right)-\frac{p^{2}}{2M}-Mv^{2}+\frac{M^{3}v^{4}}{p}\right\};\;\;\\ \varepsilon \left(p\right)={\left[v^{2}p^{2}+{\left(\frac{p^{2}}{2M}\right)}^{2}\right]}^{1/2};\;v^{2}=\frac{\sqrt{2\pi }\hbar^{2}\alpha }{M^{2}}\frac{N}{S} \end{array}$$

In this expression, $E_{0}$ is the energy of the ground state of the exciton gas (minus the particle size quantization energy) and the expression:(29)$$\varepsilon \left(p\right)={\left[v^{2}p^{2}+{\left(\frac{p^{2}}{2M}\right)}^{2}\right]}^{1/2},$$
is the energy spectrum of elementary excitations of the exciton condensate. As we can see, the spectrum of elementary excitations is determined by the momentum of the translational motion of particles and their density in the NPL plane, as well as by the dimensionless two-dimensional scattering parameter from (23).

## 3. Results and Discussions

Under conditions when the thermalization time of excitons in the sample is significantly less than their recombination time, the presence of an energy gap between the ground and higher energy states of size quantization of the movement of the center of mass of a two-dimensional exciton in the CdSe NPL plane $L_{x}\times L_{y}$ makes it possible for the BEC of two-dimensional excitons to occur in such a nanostructure (see Figure 1 and Figure 2). The condensation temperature is determined by the size quantization energy of charge carriers along the z-axis, the size quantization energy of the translational motion of the center of mass of two-dimensional excitons in the plane, and the binding energy of a two-dimensional exciton. Since all these energy quantities are determined by the corresponding geometric dimensions of the sample, the conclusion is obvious that the Bose condensation temperature of excitons $T_{c}$ is directly determined by the geometric dimensions of the sample.

Calculations carried out in the ideal Bose gas approximation clearly show (see Figure 2) that at the same particle density, with an increase in the transverse dimensions of the NPL, i.e., with a decrease in the above-mentioned energy gap between the ground and excited states of the energy of the translational motion of a two-dimensional exciton in the plane, the value of $T_{c}$ decreases. In the limit of NPL, the transition to a two-dimensional subsystem vanishes, as it should be within the limit of a two-dimensional ideal Bose gas. As expected from general considerations, at fixed sample sizes, the condensation temperature increases with increasing particle density.

The dependence of the condensation temperature $T_{c}$ on the density of particles in this consideration also reveals the following regularity: as is known, in a three-dimensional ideal Bose gas, the dependence of temperature on the number of particles N is as follows:(30)$$T_{c}^{3D}∼N^{3/2}$$

Figure 4 shows plots of the number of ideal Bose gas particles in the three-dimensional case and in the case of an NPL. As we can see, at low densities, the behavior differs significantly from the law $T_{c}^{3D}∼N^{3/2}$, while at high densities, the curves for the NPL and the bulk sample practically coincide. The latter circumstance can be explained by the fact that the NPL itself is, strictly speaking, still a three-dimensional structure and, at a high density of particles, its anomalously small dimensions in one of the directions will not have a significant effect on the statistical properties of the particles.

Indeed, with an increase in the number of particles at temperatures above the critical one, the probability of filling high quantum-well levels with particles also increases. Moreover, the filling of quantization levels along the z-axis considers the Pauli principle. But according to the quantum mechanical correspondence principle, in this case, the energy spectrum of particle motion along the z-axis will approach the semiclassical limit, i.e., becomes quasi-continuous. Considering that the center of mass motion of a two-dimensional exciton in the plane of NPL above the second level is also quasi-continuous, we can conclude that, at high densities, the statistical properties of particles in NPLs approach the statistical properties of a bulk sample. This circumstance also manifests itself in the behavior of the condensation temperature at high densities $T_{c}^{3D}∼N^{3/2}$.

Table 2 shows the values of the maximum possible (critical) values of the two-dimensional exciton density $n_{\max}^{2D},{\mathrm{cm}}^{-2}$ for various characteristics of CdSe NPLs in ideal Bose gas approximation.

As already noted, ceteris paribus, the presence of saturated screening leads to an increase in the value of the condensation temperature. Table 3 shows the values of the maximum possible (critical) values of the two-dimensional exciton density $n_{\max}^{2D},{\mathrm{cm}}^{-2}$ for various characteristics of CdSe NPLs in the case of saturated screening [41,60].

A comparison of Table 2 and Table 3 clearly shows that, in the case where screening is considered, the limiting particle density and, respectively, the BEC temperature in CdSe NPLs at the same geometric dimensions increases compared to the case where screening is absent. This is due to a decrease in the absolute value of the “internal energy” of the two-dimensional exciton due to screening [62].

When the weak nonideality of a two-dimensional exciton gas is considered, the specificity of two-dimensional scattering at low energies manifests itself due the fact that the scattering amplitude (also the scattering cross-section) increases with decreasing energy of the scattering particles. In this case, parameter (23) or an equivalent parameter:(31)β=Kσ

From expressions (23) and (31), it is clear that if, at some value of the quasi-wave vector K (or energy) of the scattering particles, a control measurement of the value of the scattering cross-section is carried out, then, according to expressions (23) and (31), it is possible to determine the value of the cross-section already at any energy values that satisfy the above requirements for low energies.

## 4. Conclusions

We now present the conclusions regarding the results obtained in the work:If the conditions when the exciton subsystem can be described in terms of its properties within the ideal Bose gas model are realized, BEC of excitons is possible in the CdSe NPL;BEC critical temperature is determined by the geometric dimensions of the sample and increases with decreasing system dimensions along the plane of the NPL; as the longitudinal dimensions of the sample tend to infinity, the BEC temperature tends to zero;At high levels of excitation, when the exciton subsystem and the subsystems of unbound electrons and holes coexist in the CdSe NPL, excitons experience a screening effect from free carriers, as a result of which the temperature of the BEC of excitons increases;BEC temperature in the presence of screening is determined by the density of excitons and non-transfer charge carriers, which is ultimately determined by the density of optical pumping;The spectrum of elementary excitations of the exciton condensate is determined by the two-dimensional density of excitons and the potential of exciton–exciton interaction.

## Figures and Tables

**Figure 1 nanomaterials-13-02734-f001:**
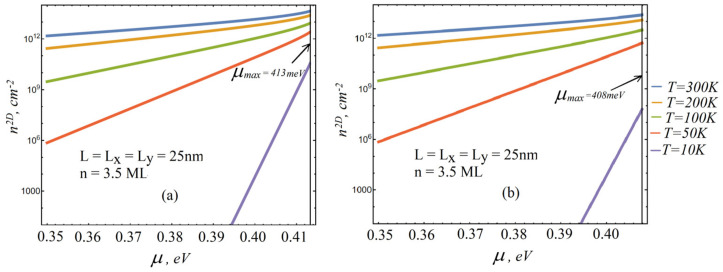
Dependence of the chemical potential μ of an equilibrium system of excitons in CdSe NPL on the two-dimensional density of particles $n^{2D}$ at various values of the system temperature T for system dimensions in the plane $L=L_{x}=L_{y}=25\mathrm{nm}$ (**a**), $L=L_{x}=L_{y}=100\mathrm{nm}$ (**b**).

**Figure 2 nanomaterials-13-02734-f002:**
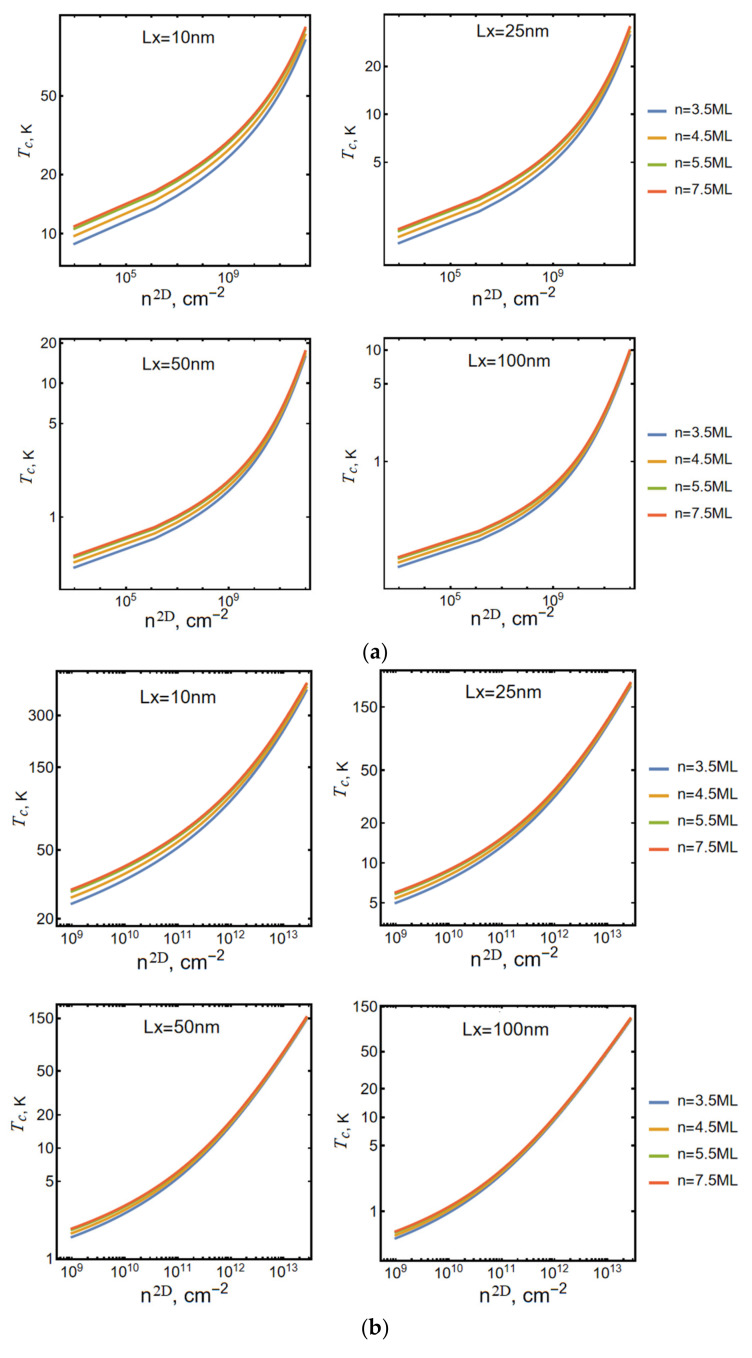
The dependence of the condensation temperature $T_{c}$ on the value of the two-dimensional particle density $n^{2D}$ at different sizes of the NPL area. (**a**) Plotted for lower values of exciton density, (**b**) larger values of density.

**Figure 3 nanomaterials-13-02734-f003:**
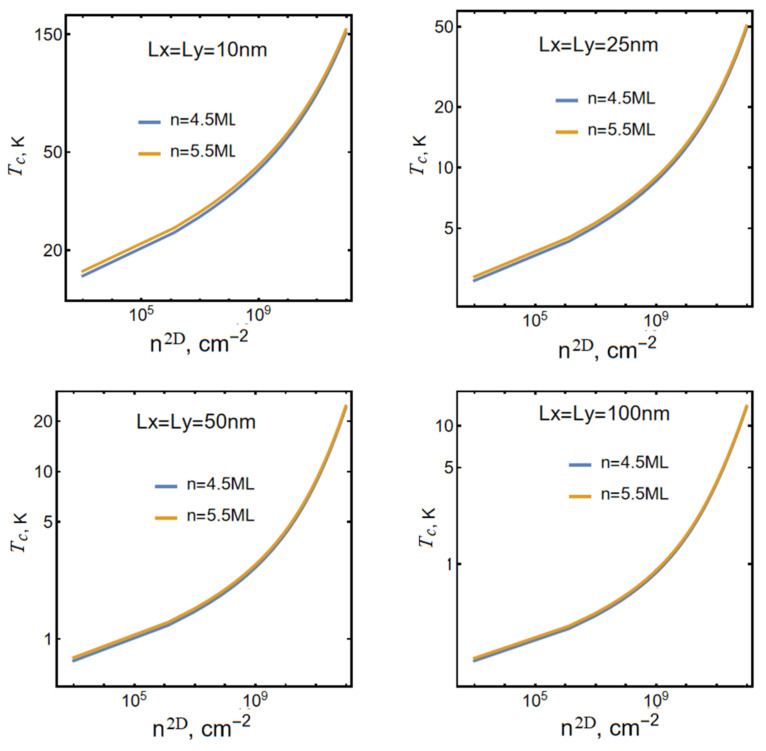
The dependence of the condensation temperature $T_{c}$ on the value of the two-dimensional particle density $n^{2D}$ at different sizes of the NPL area in the case of the presence of free charges and saturated screening. From a comparison of the graphs in Figure 2 and Figure 3, it is clearly seen that in the presence of saturated screening, under otherwise identical conditions, the temperature of the Bose condensation increases.

**Figure 4 nanomaterials-13-02734-f004:**
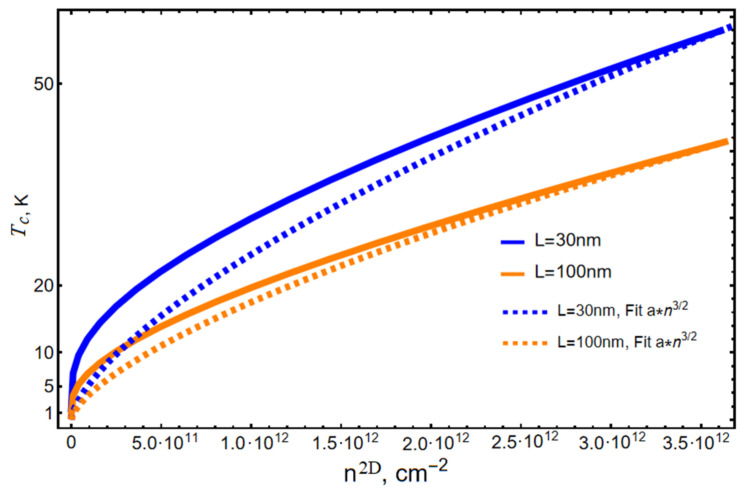
Plots of ideal Bose gas condensation temperature in the three-dimensional case (dashed lines) and in the case of an NPL (solid line).

**Table 1 nanomaterials-13-02734-t001:** Values of physical quantities characterizing the CdSe NPL under consideration, at which the graphs in Figure 1 were calculated according to expression (3). Data taken from Ref. [38].

NPL Size (nm)	NPL Layers	$m_{P}^{e}$	$m_{P}^{h}$	${\left(E_{ex,conf}^{P}\right)}_{\min}$ meV	$\begin{array}{l} {\left(E_{⊥,conf}^{e}\right)}_{\min}\\ +{\left(E_{⊥,conf}^{h}\right)}_{\min} \end{array}$ meV	${\left(E_{b}^{ex}\right)}_{\min}$ meV
$L_{x}=L_{y}=100$	3.5	0.119 $m_{0}$	0.733 $m_{0}$	0.365	720	312.8
$L_{x}=L_{y}=25$	3.5	0.119 $m_{0}$	0.733 $m_{0}$	4.06	720	312.7

Here $m_{0}$ is the free electron’s mass.

**Table 2 nanomaterials-13-02734-t002:** The maximal density of excitons $n_{\max}^{2D},{\mathrm{cm}}^{-2}$ computed by close packing of the excitons with the Bohr radius $a_{B}^{2D},\mathrm{nm}$ in the CdSe NPL plane.

n,ML	$\begin{array}{l} L=L_{x}=L_{y},\\ \mathrm{nm} \end{array}$	$n_{max}^{2D},{\mathrm{cm}}^{-2}$	$a_{B}^{2D},\mathrm{nm}$	$\mu_{max},\mathrm{eV}$	$T_{c},K$
3.5	10	1.29 × 10^13^	1.493	0.444	279.42
3.5	25	1.28 × 10^13^	1.493	0.413	129.39
3.5	50	1.28 × 10^13^	1.493	0.408	83.661
3.5	100	1.28 × 10^13^	1.493	0.407	59.470
4.5	10	9.09 × 10^12^	1.775	0.323	270.4
4.5	25	9.07 × 10^12^	1.777	0.288	107.71
4.5	50	9.07 × 10^12^	1.777	0.283	67.135
4.5	100	9.07 × 10^12^	1.777	0.282	46.43
5.5	10	6.94 × 10^12^	2.031	0.250	240.3
5.5	25	6.87 × 10^12^	2.041	0.211	94.43
5.5	50	6.87 × 10^12^	2.041	0.206	56.99
5.5	100	6.87 × 10^12^	2.041	0.204	38.46
7.5	10	5.21 × 10^12^	2.344	0.159	201.11
7.5	25	5.04 × 10^12^	2.383	0.117	80.05
7.5	50	5.04 × 10^12^	2.383	0.111	46.80
7.5	100	5.04 × 10^12^	2.383	0.110	30.81

**Table 3 nanomaterials-13-02734-t003:** Excitons in CdSe NPLs in the case of saturated screening. The maximal density of excitons $n_{\max}^{2D},{\mathrm{cm}}^{-2}$ computed by close packing of the excitons with the Bohr radius $a_{B}^{2D},\mathrm{nm}$ in the CdSe NPL plane.

n,ML	$\begin{array}{l} L=L_{x}=L_{y},\\ \mathrm{nm} \end{array}$	$n_{max}^{2D},{\mathrm{cm}}^{-2}$	$a_{B}^{2D},\mathrm{nm}$	$\mu_{max},\mathrm{eV}$	$T_{c},K$
4.5	10	1.42 × 10^13^	1.421501	0.579351	397.067
4.5	25	1.42 × 10^13^	1.421582	0.555023	195.512
4.5	50	1.42 × 10^13^	1.42158	0.551548	128.855
4.5	100	1.42 × 10^13^	1.421583	0.550679	91.6116
5.5	10	1.11 × 10^13^	1.606809	0.451801	365.676
5.5	25	1.11 × 10^13^	1.607317	0.425445	172.604
5.5	50	1.11 × 10^13^	1.607319	0.42168	110.4491
5.5	100	1.11 × 10^13^	1.607322	0.420739	88.255

## Data Availability

Not applicable.

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
