# Peer review of "Possibility of Exciton Bose–Einstein Condensation in CdSe Nanoplatelets"

_nanomaterials, 2023, doi:10.3390/nano13192734_

Round 1

Reviewer 1 Report

The manuscript entitled, ‘Possibility of Exciton Bose-Condensation in CdSe Nanoplatelets’ reported Exciton Bose-Condensation for nanoparticles. The article should be modified according to the following points;

1.      The nanomaterial used here, has fluorescent behavior. Does it affect the Exciton Bose-Condensation?

2.      Author also discussed about nanoplatelets. How that differs from quasi spherical nanoparticles especially for dots type particles?

3.      Some articles have significance and could be discussed with the help of following references:

(a)    Das, T. K., & Ganguly, S. (2023). Revolutionizing Food Safety with Quantum Dot–Polymer Nanocomposites: From Monitoring to Sensing Applications. Foods12(11), 2195.

(b)   Das, P., Ganguly, S., Banerjee, S., & Das, N. C. (2019). Graphene based emergent nanolights: A short review on the synthesis, properties and application. Research on Chemical Intermediates45, 3823-3853.

(c)    Xu, Q., Niu, Y., Li, J., Yang, Z., Gao, J., Ding, L., ... & Xu, C. (2022). Recent progress of quantum dots for energy storage applications. Carbon Neutrality1(1), 13.

Author Response

We express our gratitude to the reviewers for a detailed review of our manuscript and useful remarks. We made the correction according to the reviewers’ comments and additionally improved the introduction and results sections as well as the English level of the manuscript. We also have done stylistic checks. See the response to revivers questions below.

Response to reviewers:

Rev.1

Тhe manuscript entitled, ‘Possibility of Exciton Bose-Condensation in CdSe Nanoplatelets’ reported Exciton Bose-Condensation for nanoparticles. The article should be modified according to the following points;

  1. The nanomaterial used here, has fluorescent behavior. Does it affect the Exciton Bose-Condensation?

Response:

We thank reviewer 1 for this comment we added the notes in the manuscript text. The Bose-Einstein condensation state is statistically equilibrium. In order to observe fluorescence in such a system, additional exposure to light radiation is necessary, which will lead to the transition of the particle`s lowest energy level to excited levels. After the cessation of exposure to external radiation, the particle will descend to the first excited level, and from there, - to the ground state level, while emitting a certain energy. Such a process is not considered within the scope of our paper. Additionally, the presence of a macroscopically large number of excitons at the ground level, i.e. in the bose-condensate ( ), during the phenomena of both stimulated and spontaneous excitonic annihilation will lead to the appearance (at the threshold frequency ) of a sharp delta-shaped and high-intensity peak ( ).

  1. Author also discussed about nanoplatelets. How that differs from quasi spherical nanoparticles especially for dots type particles?

We thank reviewer 1 for this comment, we have added a paragraph about NPL properties and particularities in the introduction. The significant advantage of NPLs is the perfection of geometric shapes and geometric shape variations (especially in the z direction where the confinement is much stronger than in lateral directions) compared to core-shell QD. In addition to what has been said, it should be noted that in NPLs dielectric effects are clearly manifested due to the presence of a dielectric environment, as a result, the NPL binding energies reach large values compared to the quantum well and larger quantum dots. Therefore, the exciton subsystem becomes more stable.

  1. Some articles have significance and could be discussed with the help of following references:

(a)   Das, T. K., & Ganguly, S. (2023). Revolutionizing Food Safety with Quantum Dot–Polymer Nanocomposites: From Monitoring to Sensing Applications. Foods12(11), 2195.

(b)  Das, P., Ganguly, S., Banerjee, S., & Das, N. C. (2019). Graphene based emergent nanolights: A short review on the synthesis, properties and application. Research on Chemical Intermediates45, 3823-3853.

(c)   Xu, Q., Niu, Y., Li, J., Yang, Z., Gao, J., Ding, L., ... & Xu, C. (2022). Recent progress of quantum dots for energy storage applications. Carbon Neutrality1(1), 13.

We have added these references in the introduction and changed the text of the citation. 

Thank you,

Hayk Sarkisyan

Reviewer 2 Report

The authors analyzed the possibility of exciton Bose-Einstein condensation for CdSe noplatelets. If the authors revise the manuscript, it is helpful for understanding the mechanism of Bose-Einstein condensation for the considered material. I recommend the following revisions:

1. Title

Bose-Condensation   ->   Bose-Einstein Condensation

2. Line 12 in Sec. 1

[old] Therefore, semiconductor NPLs are very promising materials that can be used to create several optoelectronic devices - light emitter devices of various ranges, light generation and amplification devices, lasers, solar energy harvesting applications, photodetectors, photosensors, photocatalysis, etc. (see e.g. Refs. [XX-XX] and references therein).

[new] Therefore, semiconductor NPLs are very promising materials that can be used to create several optoelectronic devices - light emitter devices of various ranges [XX], light generation and amplification devices [XX], lasers [XX], solar energy harvesting applications [XX], photodetectors [XX], photosensors [XX], photocatalysis [XX], etc.

3. The authors represented explanations of subsequent sections in the last paragraph of Sec. 1. However, they are not match with the contents of actual sections.

4. Equation 1 includes a_{ex}^{3D}. However, the authors did not represent what it is.

5. After Eq. 2 (and line 6 in “Results and Discussions” section)

the Z axis  ->  the z axis

6. Before Eq. 6

The symbol of temperature T should be an italic letter.

7. After Eq. 6

The meaning of the word “majorant” may be obscure.

8. Equation 9 includes E_{n_e,n_h}^{conf}. However, the authors did not represent the definition of E_{n_e,n_h}^{conf}. Although the authors represented the definitions of E_{n_e}^{conf} and E_{n_h}^{conf}, they should represent the definition of E_{n_e,n_h}^{conf} separately for clarity.

9. In table 1, I recommend to revise as:

m_{||}^e,m_0  ->  m_{||}^e

m_{||}^h,m_0  ->  m_{||}^h

0.1198  ->  0.1198m_0

0.733  ->  0.733m_0

10. After table 1

0,408eV  ->  0.408eV

0,413eV  ->  0.413eV

11. Figure 2

The sizes of panels in Figure 2 are not uniform. The authors should revise this.

In addition, n2d in figure 2 should be revised as;

n2d  ->  n^{2D} 

12. On page 8, the phrase “when the exciton binding energy is practically is,” should be revised.

13. In Figure 3

n2d  ->  n^{2D}

14. After Eq. 20

where q – transmitted   ->  where q is transmitted

15. After Eq. (27)

In this expression, is the energy  ->  In this expression, E_0 is the energy

16. Line 5 in “Results and Discussions” section

see Fig. 1, 2   ->   see Figs. 1 and 2

17. Table 2

n_{2d}^{max}     ->    n_{max}^{2D}

18. Before Eq. (29)

The symbol N should be an italic letter.

END

Author Response

We express our gratitude to the reviewers for a detailed review of our manuscript and useful remarks. We made the correction according to the reviewers’ comments and additionally improved the introduction and results sections as well as the English level of the manuscript. We also have done stylistic checks. See the response to revivers questions below.

Response to reviewers:

Rev.2

The authors analyzed the possibility of exciton Bose-Einstein condensation for CdSe noplatelets. If the authors revise the manuscript, it is helpful for understanding the mechanism of Bose-Einstein condensation for the considered material. I recommend the following revisions:

  1. Title: Bose-Condensation   ->   Bose-Einstein Condensation

Response: We thank reviewer 2 for this comment. We made the change according to the reviewer 2 comment, now the Title change to  “POSSIBILITY of BOSE-EINSTEIN CONDENSATION in CdSe NANOPLATELETS”

  1. 2. Line 12 in Sec. 1

[old] Therefore, semiconductor NPLs are very promising materials that can be used to create several optoelectronic devices - light emitter devices of various ranges, light generation and amplification devices, lasers, solar energy harvesting applications, photodetectors, photosensors, photocatalysis, etc. (see e.g. Refs. [XX-XX] and references therein).

[new] Therefore, semiconductor NPLs are very promising materials that can be used to create several optoelectronic devices - light emitter devices of various ranges [XX], light generation and amplification devices [XX], lasers [XX], solar energy harvesting applications [XX], photodetectors [XX], photosensors [XX], photocatalysis [XX], etc.

Response: We thank reviewer 2 for this comment. We made the change according to the reviewer 2 comment:

Therefore, semiconductor NPLs are very promising materials that can be used to create several optoelectronic devices - light emitter devices of various ranges [XX], light generation and amplification devices [12,16], lasers [16-18], solar energy harvesting applications [19-20], photodetectors [21-24], photosensors [22-24], etc.

  1. The authors represented explanations of subsequent sections in the last paragraph of Sec. 1. However, they are not match with the contents of actual sections.

Response:  We thank reviewer 2 for this comment. We made the change according to the reviewer 2 comment:

In Section 2 the theoretical approach for Bose-Einstein condensation of excitons in the frameworks of ideal (sub-section 2.1, 2.2) and weakly non-ideal bose gas (sub-section 2.3) is presented. In Section 3 the results, obtained in the work are discussed and in Section 4 the conclusion related to the work are presented. One APPENDIX is present also in the work.

  1. Equation 1 includes a_{ex}^{3D}. However, the authors did not represent what it is.

Response: We thank reviewer 2 for this comment. We made the change according to the reviewer 2 comment. Before Eq.1 it is additionally represented, that a_{ex}^{3D} is the bohr radius of exciton in bulk CdSe.

  1. After Eq. 2 (and line 6 in “Results and Discussions” section)

the Z axis  ->  the z axis

Response: We thank reviewer 2 for this comment. We made the change according to the reviewer 2 comment. Necessary changes to the text have been made

  1. Before Eq. 6

The symbol of temperature T should be an italic letter.

Response: We thank reviewer 2 for this comment. We made the change according to the reviewer 2 comment. Necessary changes to the text have been made

  1. After Eq. 6 The meaning of the word “majorant” may be obscure.

Response: We thank reviewer 2 for this comment. We made the change according to the reviewer 2 comment. We modified this sentence.

  1. 8. Equation 9 includes E_{n_e,n_h}^{conf}. However, the authors did not represent the definition of E_{n_e,n_h}^{conf}. Although the authors represented the definitions of E_{n_e}^{conf} and E_{n_h}^{conf}, they should represent the definition of E_{n_e,n_h}^{conf} separately for clarity.

Response: We thank reviewer 2 for this comment. We made the change according to the reviewer 2 comment. Here  is the total energy of size quantization of an electron and a hole along the z axis.

  1. In table 1, I recommend to revise as:

m_{||}^e,m_0  ->  m_{||}^e

m_{||}^h,m_0  ->  m_{||}^h

0.1198  ->  0.1198m_0

0.733  ->  0.733m_0

10 After table 1

0,408eV  ->  0.408eV

0,413eV  ->  0.413eV

Response: We thank reviewer 2 for this comment. We made the change according to the reviewer 2 comment in table 2 and the text below.

  1. Figure 2 The sizes of panels in Figure 2 are not uniform. The authors should revise this. In addition, n2d in figure 2 should be revised as; n2d -> n^{2D}

Response: We thank reviewer 2 for this comment. We made the change according to the reviewer 2 comment.

  1. On page 8, the phrase “when the exciton binding energy is practically is,” should be revised.

Response: We thank reviewer 2 for this comment. We made the change according to the reviewer 2 comment. There was a typo the corrected text is “when the exciton binding energy is practically is constant”

  1. In Figure 3 n2d  ->  n^{2D}

Response: We thank reviewer 2 for this comment. We made the change according to the comment.

  1. After Eq. 20 where q – transmitted   -> where q is transmitted

Response: We thank reviewer 2 for this comment. We made the change according to the comment.

  1. After Eq. (27) In this expression, is the energy -> In this expression, E_0 is the energy

Response: We thank reviewer 2 for this comment. We made the change according the comment.

  1. 16. Line 5 in “Results and Discussions” section see Fig. 1, 2   ->   see Figs. 1 and 2

Response: We thank reviewer 2 for this comment. We made the change according to the comment.

  1. Table 2 n_{2d}^{max}     ->    n_{max}^{2D}

Response: We thank reviewer 2 for this comment. Its was a type and it has been fixed.

  1. Before Eq. (29) The symbol N should be an italic letter.

Response: We thank reviewer 2 for this comment. We made the change according to the reviewer 2`s comment.

Thank you,

Hayk Sarkisyan

Round 2

Reviewer 1 Report

This can be published in its present form. 

Reviewer 2 Report

The authors revised the manuscript according to my report. Now I recommend the publication of this manuscript in Nanomaterials.